# Effects of polyethylene glycol 2 L alone or with ascorbic acid compared with polyethylene glycol 4 L alone for bowel preparation before colonoscopy: protocol for a systematic review and network meta-analysis

Xu Tian,[1,2] Wei-Qing Chen,[1] Jie-Li Huang,[1] Lan-Ying He,[1] Bang-Lun Liu,[1] Xi Liu,[1] Hang Zhou,[1] Bing-Rong Liu[3]

XT and W-QC contributed equally.

[1]Department of Gastroenterology, Chongqing Cancer Institute and Hospital and Cancer Centre, Chongqing, China
[2]Editorial Office, TMR Integrative Nursing, TMR Group, Tianjin, China
[3]Gastrointestinal Hospital, First Affiliated Hospital of Zhengzhou University, Zhengzhou, China

**Correspondence to**
Professor Wei-Qing Chen;
chenwq20140809@163.com

## ABSTRACT

**Introduction** Colonoscopy has been regarded as a standard method of detecting and removing gastrointestinal lesions early, while adequate bowel preparation is the prerequisite of determining the diagnostic accuracy and treatment safety of this process. Polyethylene glycol (PEG) based bowel preparation regimens remain the first recommendation, but the optimal option is still uncertain. The aim of this systematic review and network meta-analysis of randomised controlled trials (RCTs) is to determine the optimal PEG based bowel preparation regimen before colonoscopy.

**Methods and analysis** We will assign two investigators to independently search all potential citations, screen records, abstract essential information and appraise the risk of bias accordingly. Then, random effects pairwise and network meta-analyses of RCTs comparing PEG 2 L alone or with ascorbic acid with PEG 4 L alone will be performed using RevMan 5.3 (Copenhagen, Denmark: The Nordic Cochrane Centre, The Cochrane Collaboration, 2013), Stata 14 (StataCorp, Texas, USA) and WinBUGS 1.4 (Imperial College School of Medicine, St Mary's, London, UK) from January 2000 to April 2017. The surface under the cumulative ranking curve will also be calculated in order to rank the regimens.

**Ethics and dissemination** Ethics approval and patient written informed consent will not be required because all of the analyses in the present study will be performed based on data from published studies. We will submit our systematic review and network meta-analysis to a peer reviewed scientific journal for publication.

**Systematic review registration** PROSPERO: CRD42017068957.

## Strengths and limitations of this study

► The protocol addresses the important question of whether polyethylene glycol (PEG) 2 L alone or with ascorbic acid compared with PEG 4 L alone offers the most benefits for bowel preparation before colonoscopy.

► The present network meta-analysis has a clearly established aim, stringent inclusion criteria, state of the art methods for data collection and quantitative synthesis.

► The present network meta-analysis will design a series of established methods to increase the reliability of the pooled results through rationally addressing heterogeneity and risk of bias.

► The present network meta-analysis will rank all investigated PEG based bowel preparation regimens in terms of each outcome, which facilitates evidence informed decision making.

► Limitations include variations in administration times of drinking the same bowel preparation regimens, diet description prior to colonoscopy, type of colonoscopies and assessment tool for bowel preparation efficacy.

method for early detection and prevention of CRC.[2] Published evidence suggested that early detection and endoscopic resection of polyps and abnormal lesions in the gastrointestinal tract can reduce mortality of CRC by approximately 50%.[3 4] However, adequate bowel preparation is a prerequisite for guaranteeing diagnostic accuracy and therapeutic safety of colonoscopy.[5] More than 40% of colonoscopy failures were a result of inadequate bowel preparation.[6] Moreover, inadequate bowel preparation also caused other negative consequences, such as missed detection of polyps or lesions, an increased

## BACKGROUND

Colorectal cancer (CRC) is one of the most common cancers diagnosed worldwide and is also a major contributor to cancer associated morbidity and mortality.[1] Colonoscopy has been considered the most effective

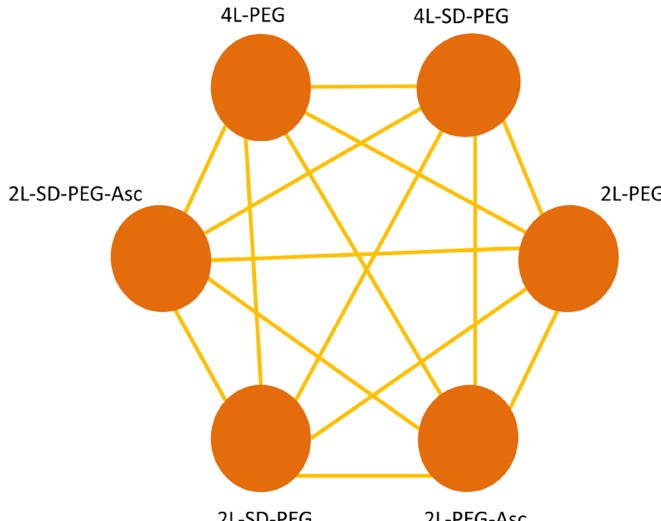

4L-PEG  4L-SD-PEG

2L-SD-PEG-Asc  2L-PEG

2L-SD-PEG  2L-PEG-Asc

**Figure 1** Possible evidence network of all possible polyethylene glycol (PEG) based bowel preparation regimens in terms of bowel preparation efficacy. The yellow solid line indicates direct comparisons between regimens which were directly compared in original studies. The brown node represents each PEG based bowel preparation regimen. Asc, ascorbic acid; SD, split dose.

risk of procedure related complications and increased economic costs.[7] Several factors can affect the quality of bowel preparation,[8] and low patient based compliance, poor palatability of the bowel preparation solution and inevitable requirement of drinking a large volume of preparation solution account for 20–25% of inadequate bowel preparations.[7] However, low patient based compliance with the recommended regimen plays a decisive role in the overall success of the procedure.[9]

For the purpose of improving the quality of bowel preparation, several regimens have been developed, such as polyethylene glycol (PEG) based solutions, sodium phosphate and sodium picosulfate solutions. Of these, PEG based regimens are the first recommendation.[10] Several modified regimens, including split dose regimens, low volume regimens and low volume plus ascorbic acid (Asc) have been designed because patients find it difficult to consume the traditional 4L PEG regimen owing to the large volume of fluid and poor palatability.[11] A series of randomised controlled trials (RCTs) have been performed to investigate the comparative efficacy of split dose versus single dose,[12] low volume (2L) plus Asc versus traditional volume (4L)[13] and low volume plus Asc versus low volume.[14] However, studies on low volume versus traditional volume, low volume versus low volume plus Asc with split dose, and low volume versus traditional volume with split dose have not been identified. Moreover, in individual studies it is difficult to identify subtle clinical differences owing to the smaller patient numbers.[15] Several meta-analyses have also been performed to evaluate the efficacy of low volume versus traditional volume,[16] low volume versus plus Asc versus traditional volume,[17] and split dose versus single dose.[18 19]

Traditional meta-analysis methods, however, are unable to investigate the comparative efficacy of more than two interventions.

In order to solve the limitations of the traditional meta-analysis technique, Bayesian network meta-analysis based on Markov Chain Monte Carlo and Gibbs Sampling, the expansion of pairwise meta-analysis, has been developed to evaluate the comparative efficacy of multiple treatments which are not directly compared in individual RCTs.[20] Thus we proposed this network meta-analysis to establish the effects of PEG 2 L alone or with Asc compared with PEG 4 L alone prior to colonoscopy. We designed this systematic review and network meta-analysis on 10 May 2017 and we expected to complete this study by 31 December 2017.

## METHODS AND DESIGN

We designed and completed this protocol for a systematic review and network meta-analysis according to the preferred reporting items for systematic reviews and meta-analysis protocols (PRISMA-P) 2015: elaboration and explanation.[21] The systematic review and network meta-analysis was registered in the International Prospective Register of Systematic Reviews (PROSPERO) (CRD42017068957). We will perform this traditional pairwise and network meta-analysis in accordance with the Cochrane Handbook for Systematic Reviews of Interventions[22] and report all results according to the preferred reporting items for systematic review and meta-analysis for network meta-analysis (PRISMA-NMA).[23]

### Selection criteria

In our meta-analysis, a study will be considered if the following inclusion criteria are met: (i) patients: all adult patients undergoing elective colonoscopy at an endoscopy centre, irrespective of whether they are outpatients and inpatients; (ii) intervention: all PEG based bowel preparation regimens, including 4L PEG and 2L PEG plus Asc with a single or split dose and not combined with other drugs—we will drawn the possible evidence network according to the targeted regimens in terms of bowel preparation efficacy (see figure 1); (iii) outcomes: bowel preparation efficacy is regarded as the primary outcome, and secondary outcomes include compliance with the recommend regimen (CP), preference to repeat the same regimen (PRSR), acceptance of the regimen (AT), adverse events (AEs) and detection rate of polyps and adenomas (DRPA) and colorectal cancer (DRCRC); (iv) study design: only RCTs will be included—an abstract with sufficient data will also be considered; and (vi) language: only full text published in the English or Chinese language will be considered, because translators well versed in other languages are not included.

A study will be excluded if it meets at least one of the following criteria: (i) essential information cannot be extracted; (ii) duplication with poor methodology and insufficient data; (iii) non-original research types, such as

review, editorial, letter to the editor or comments; and (iv) a study investigating bowel preparation regimen in special patients, such as the elderly or in patients with a previous poor bowel preparation.

## Definition of outcomes

In our systemic review and network meta-analysis, bowel preparation efficacy is also regarded as successful bowel preparation, and is defined as an Ottawa score of <5, a Boston Bowel Preparation Scale score of ≥2 for all locations, an excellent or good bowel preparation designation on the Aronchik Scale, or other non-validated 3, 4 or 5 point scales (excellent, good, fair, poor, very poor) rated by the colonoscopist while performing the colonoscopy. CP was defined as adherence to the bowel preparation prescribed or consumption of at least 75% of the prescribed bowel preparation, which was evaluated before the colonoscopy was performed. PRSR, AT and AEs were measured using the specified questionnaires in each eligible study after completion of the colonoscopy examination (ie, defined by the individual study). DRPA and DRCRC refer to the number of polyps and adenomas actually detected and CRC, respectively, which were all established histopathologically.

## Identification of citations

We will electronically search PubMed, Cochrane Central Register of Controlled Trials (CENTRAL), EMBASE, China National Knowledge Infrastructure (CNKI) and Chinese Biomedical Literatures database (CBM) in order to capture all potential records investigating the comparative efficacy of different PEG based bowel preparation regimens from January 2000 to April 2017. 'Colonoscopy', 'polyethylene glycols' and 'random' will be used to construct search algorithms in accordance with the requests of targeted databases, and all possible search algorithms have been documented in the online supplementary material table 1.

After the electronic searches, we will also hand check the reference lists of all eligible studies and topic related reviews and electronically retrieve the Clinicaltrial. gov for the purpose of covering all potential eligible studies. However, only studies published in English and Chinese will be considered in our systematic review and network meta-analysis.

## Data extraction

We have designed a standard data extraction form, used in our previous two systematic reviews and network meta-analyses (see online supplementary material-SDE). All captured citations will be imported into EndNote literature management software V.X7. We will then assign two reviewers to abstract the basic information and data for the specific outcomes from the eligible studies, such as first author, publication year, age of participants, sample size, bowel preparation regimens and outcomes of interest using this standard data extraction form.[24] We will contact the corresponding author if sufficient data of an eligible study cannot be abstracted from the full text. The kappa value will be calculated to assess inter-investigator reliability. We will establish the consensus principle as the method of resolving differences between reviewers.

## Quality assessment of individual study

We will assign two independent reviewers to appraise the risk of bias from seven domains, including randomisation sequence generation, allocation concealment, blinding of participants, blinding of study personnel, blinding of outcome assessors, incomplete outcome data, selective reporting and other bias with the Cochrane risk of bias assessment tool.[22 25] A study will be assigned a risk level of 'high risk of bias', 'unclear risk of bias' or 'low risk of bias' according to the match level between the actual information and the evaluation criteria.[22]

## Description of the available data

We will derive each pairwise comparison from descriptive statistics on the available data and selected variables for the study and population characteristics, such as age, study length and outcome relevant baseline risk factors. A network diagram will be used for each outcome to present the direct comparisons between the different bowel preparation regimens and control groups. In these diagrams, nodes (circles) represent various bowel preparations and their sizes are proportional to the sample size of each respective intervention; edges (lines) indicate direct comparisons and their thickness is proportional to the standard error (precision).

## Statistical analysis

We will first perform a traditional pairwise meta-analysis based on the random effect model, which incorporates within and between studies heterogeneity, to estimate the summarised OR and 95% CIs.[26] The $\chi^2$ method will be adopted to test the heterogeneity[27] and the $I^2$ statistic will be used to estimate the proportion of the overall variation that is attributable to between study heterogeneity.[28] A value for the $I^2$ statistic >50% indicates substantial heterogeneity.[28] We will draw the funnel plot to identify publication bias if the number of studies analysed is more than 10.[29] The studies with more than two comparison groups will be quantitatively incorporated into the pairwise meta-analysis according to the specific comparison.

Following the traditional pairwise meta-analysis, a random effects network meta-analysis will be performed according to the methods described by Chaimani et al.[30] The initial values, automatically generated from the software, will be used to fit the model.[31] We plan to perform 70 000 iterations and 30 000 burn-in for each outcome and convergence.

The surface under the cumulative ranking curve (SUCRA) will also be drawn to rank all PEG based bowel preparation regimens, with a higher value suggesting better results for the respective regimen.[32]

All analyses will be conducted using the RevMan 5.3 (Copenhagen, Denmark: The Nordic Cochrane Centre,

The Cochrane Collaboration, 2013), Stata 14 (StataCorp, Texas, USA) and WinBUGS 1.4 (Imperial College School of Medicine, St Mary's, London, UK).

### Assessment of small study effects and inconsistency

We will generate the comparison adjusted funnel plot to assess the small study effects when the number of studies included in one pair of comparison is more than 10.[33] We will calculate the inconsistency factor based on the loop specific method to assess the inconsistency.[30]

### Subgroup and sensitivity analyses

In the case of possible important heterogeneity or inconsistency, we will explore the possible sources using subgroup and meta-regression analyses. Subgroup analyses are planned for time of colonoscopy, patient sources and age. Sensitivity analyses are planned for bowel preparation quality by analysing only studies considered at low risk of bias.

## DISCUSSION

CRC is one of the most common malignancies, and statistics indicate that it is the fourth contributor to cancer death worldwide.[1] Colonoscopy has been regarded as the standard process for early prevention and detection of CRC in clinical practice.[2] However, diagnostic accuracy and operation safety while performing colonoscopy mainly depend on the quality of bowel preparation.[34] Although several novel bowel preparation regimens have been developed to improve the tolerability and compliance of patients, PEG based regimens remain the firstline recommendation.[10] Several modified regimens have been applied in clinical practice, but no primary study or traditional pairwise meta-analysis comparing various PEG based bowel preparation regimens has been published. Thus it is still unclear which PEG based regimen is optimal. We have proposed a network meta-analysis to determine the optimal PEG based regimen for the purpose of facilitating the informed decision making process.

This network meta-analysis will be one of the first to compare the direct and indirect effects of different PEG based regimens for bowel preparation prior to colonoscopy. The results of the network meta-analysis will influence evidence based decision making for bowel preparation regimen prescriptions as it will be fundamental in providing reliable recommendations for bowel preparation regimens before colonoscopy.

### Ethics and dissemination

Ethics approval and patient written informed consent will not be required because all analyses in the present study will be performed based on data from published studies. We will submit our systematic review and network meta-analysis to a peer reviewed scientific journal for publication.

**Contributors** XT and W-QC conceived and designed this study. XT and J-LH searched and selected the studies. L-YH and B-LL extracted the essential information. XT and B-RL assessed the risk of bias. XT, W-QC, XL and HZ performed the statistical analyses. XT and W-QC interpreted the pooled results. XT, W-QC and B-RL drafted the manuscript. All authors approved the manuscript to be considered for publication.

**Competing interests** None declared.

**Provenance and peer review** Not commissioned; externally peer reviewed.

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
