## [Reviewer comments · BMJ Open]

ARTICLE DETAILS

TITLE (PROVISIONAL)	Effects of comparing 2 liters polyethylene glycol alone or plus ascorbic acid and 4 liters polyethylene glycol alone with each other for bowel preparation before colonoscopy: protocol for a systematic review and network meta-analysis
AUTHORS	Tian, Xu; Chen, Wei-Qing; Huang, Jie-Li; He, Lan-Ying; Liu, Bang-Lun; Liu, Xi; Zhou, Hang; Liu, Bing-Rong

VERSION 1 – REVIEW

REVIEWER	Susan Hutfless Johns Hopkins University, USA
REVIEW RETURNED	07-Aug-2017

GENERAL COMMENTS	The Methods section is written in the past tense in some places implying that the study is complete. The search string is not clearly provided although the study has been underway since May and will be complete in September. For example, under search id#1 for CENTRAL, I got 0 results when I tried to test the search. For search #8, I got only 332 hits. It was unclear to me why the word "random" is required in a clinical trial of prep methods as is required by these searches. Are the authors doing a review of previously published systematic reviews only or are they searching for studies de novo? It is not possible to assess if this extremely short timeframe is reasonable to conduct the study because the databases searched and the strings used are not provided transparently and with clear rationale for the terms selected. The outcomes are not specified with respect to timing or how studies that do not use the measures will be included. In the USA the Boston prep score is used. Will all USA studies be excluded? The important outcome of colorectal cancer is not included. Adenoma detection is not captured either. PEG is not compared with other bowel prep methods. How will a network meta-analysis help to identify the best prep method if only a single prep method is included? The exclusion criteria are too vague to evaluate. Please share the data extraction forms and the name of the software tool used to manage the dual independent abstraction.
---

	What is the plan for quantifying reviewer concordance at the title, abstract, full, text data abstraction phases (inter/intra-rater reliability)? How will low concordance/disagreements be handled and documented (i.e., define and quantify "consensus principle"). How will results differ from the ongoing Cochrane review that compares all methods for bowel prep? Protocol available at: http://onlinelibrary.wiley.com/doi/10.1002/14651858.CD006330.pub2/full
--	---

REVIEWER	Zhizheng Ge Division of Gastroenterology and Hepatology, Renji Hospital, School of Medicine , Shanghai Jiaotong University, Shanghai Institute of Digestive Disease Shanghai, China
REVIEW RETURNED	20-Aug-2017

GENERAL COMMENTS	Overall this is an interesting and well-designed manuscript of great practical utility. Several comments should be modified before it can be accepted for publication.  1. The topic is PEG based regimes for bowel preparation prior to colonoscopy and I wonder whether there were only 6 possible nets in this NMA? How to standardize different dosage of PEG (by grams or liters)? I also found the use of 1L PEG in several studies and I suggest regrouping the usage of PEG combined time with dosage. 2. The manuscript was of great arrangement and followed the PRISMA-P. You mentioned that this protocol was designed on May 10, 2017 and I wonder the update progress of this trial. Please register this study. 3. In the Identification of citations (Page 8 Line 130), you noted that only studies in English will be considered in this analysis. Please add this item in Selection criteria. 4. Several studies compared the bowel preparation regimes in special patients (elderly, inpatient or previous poor bowel preparation), which might influence the choice of regime. I would like to know whether you include these articles. Please specify it in your manuscript. 5. In Data extraction part, you list basic information that abstracted by 2 reviewers. The influence factors of the stratification analysis need to be added to explore its effect on the results (as age, time of colonoscopy and race)thus increasing the credibility of the conclusion.Please introduce the layered analysis method and methods to reduce bias in detail. 6. In clinical practice, good bowel preparation is for better detection of lesions. Please increase the detection rate of polyps and adenomas as one of the endpoint.
---

VERSION 1 – AUTHOR RESPONSE

Comments from Reviewer #1

Comment: Please share the data extraction forms and the name of the software tool used to manage the dual independent abstraction. What is the plan for quantifying reviewer concordance at the title, abstract, full, text data abstraction phases (inter/intra-rater reliability)? How will low concordance/disagreements be handled and documented (i.e., define and quantify "consensus principle").

Response to comment: Dear reviewer, we must express our warm appreciation to you due to your valuable comments. We uploaded the data extraction form in submission system, and introduced the software of managing the dual independent abstraction. (Please see line 166 to 168 in page 8). We also added the information of introducing the methods of quantifying reviewer concordance and handling and documenting concordance (please see line 173 to 175 in page 9).

Comment: How will results differ from the ongoing Cochrane review that compares all methods for bowel prep? Protocol available at:
<http://onlinelibrary.wiley.com/doi/10.1002/14651858.CD006330.pub2/full>

Response to comment: Dear reviewer, thanks for you information of providing the link to this Cochrane protocol to me. Our team carefully reviewed this protocol before designed our systematic review. In the given Cochrane protocol, these authors primarily determine the colonic cleanliness by using among the various agents and their side effects, including PEG, NaP, SPS, and combinations of different agents. As a result, four comparisons were designed as following:

- (1) PEG compared with NaP
- (2) PEG compared with SPS
- (3) NaP compared with SPS
- (4) combinations of different agents

However, in our present study, the primary aim is to determine the comparative efficacy of various modified PEG-based preparations, including 2L PEG alone with single or split dose, 4L PEG alone with single or split dose, 2L PEG plus Asc with single or split dose. Moreover, in that Cochrane protocol, a subgroup analysis based on various modified PEG regimes was not designed. And thus, it is rational and there is an urgent need to develop our study.

Comments from Reviewer #2 Professor Zhizheng Ge (Renji Hospital):

1. The topic is PEG based regimes for bowel preparation prior to colonoscopy and I wonder whether there were only 6 possible nets in this NMA? How to standardize different dosage of PEG (by grams or litters)? I also found the use of 1L PEG in several studies and I suggest regrouping the usage of PEG combined time with dosage.

Response to comment: Dear referee, we appreciate you for reviewing our manuscript and proposing pertinent comments.

In fact, the aim of designing this study is to systematically investigate the effect of comparing 2 liters polyethylene glycol alone, 2 liters polyethylene glycol alone plus ascorbic acid and 4 liters polyethylene glycol alone, which are all prescribed by single or split dose, each other for bowel preparation prior to colonoscopy, and thus only 6 possible nets will be founded in the study.

In the present study, we will standardize different dosage of PEG by litters.

As explained above, we design this study aiming to comparing 2 liters polyethylene glycol alone, 2 liters polyethylene glycol alone plus ascorbic acid and 4 liters polyethylene glycol alone with single or split dose each other for bowel preparation and thus other PEG-based regimes won't be considered such as 3, 1.5 or 1 liter PEG-based regime.

Moreover, as one example, based on the captured results using our designed search strings, three studies investigated the comparative efficacy of 2L PEG plus Asc versus 1L PEG plus Asc plus Bisacodyl. In these 3 studies, Bisacodyl was prescribed to combine with 1L PEG, which was also ineligible for our criteria.

2. The manuscript was of great arrangement and followed the PRISMA-P. You mentioned that this protocol was designed on May 10, 2017 and I wonder the update progress of this trial. Please register this study.

Response to comment: Dear Professor Ge, you would be warmly appreciated due to this valuable comment proposed by you. According to your suggestion, we have registered this systematic review in International Prospective Register of Systematic Reviews (PROSPERO) and a number of CRD42017068957 has been assigned (Please see line 112 to 114 in Page 6). The registered protocol can be obtained through browsing the https://www.crd.york.ac.uk/PROSPERO/register_new_review.asp

3. In the Identification of citations (Page 8 Line 130), you noted that only studies in English will be considered in this analysis. Please add this item in Selection criteria.

Response to comment: Dear reviewer, thanks for your comment. The Publication Language has been added to Selection criteria section. This information can be found in Line 131 to 132 in Page 7.

4. Several studies compared the bowel preparation regimes in special patients (elderly, inpatient or previous poor bowel preparation), which might influence the choice of regime. I would like to know whether you include these articles. Please specify it in your manuscript.

Response to comment: Dear reviewer, in the Selection section, we stated that all adult patients undergoing elective colonoscopy irrespective of outpatients and inpatients will be considered. However, we won't consider other special patients, for example elderly or previous poor bowel preparation. This information has been added in Exclusion criteria section. (Please see line 136 to 138 in Page 7)

5. In Data extraction part, you list basic information that abstracted by 2 reviewers. The influence factors of the stratification analysis need to be added to explore its effect on the results (as age, time of colonoscopy and race) thus increasing the credibility of the conclusion. Please introduce the layered analysis method and methods to reduce bias in detail.

Response to comment: Dear referee, thanks for your constructive suggestions. In our revised protocol, the subgroup analyses have been designed, which will be carried out based on the time of colonoscopy, patient sources and age (please see line 228 to 233 in page 11). Moreover, we also will perform sensitivity analysis through only including eligible studies with low risk of bias (please see line 228 to 233 in page 11).

6. In clinical practice, good bowel preparation is for better detection of lesions. Please increase the detection rate of polyps and adenomas as one of the endpoint.

Response to comment: We must express our warm appreciation to this reviewer due to this greatly valuable comment. According to the comments, we revised the outcomes which will be evaluated in our systematic review (included the detection rate of polyps and adenomas). Please see Line 130 of Page 7.

We sincerely invite all previous reviewers to review our revised manuscript if it is convenient for them.

I'm looking forward to receiving reply on final decision of my manuscript from you and anonymous peer reviewers as soon as possible.

VERSION 2 – REVIEW

REVIEWER	Susan Hutfless Johns Hopkins University, USA no new competing interests
REVIEW RETURNED	05-Sep-2017

GENERAL COMMENTS	The authors state that they revised the study to include cancer outcomes but no such changes exist in the track changed version of the draft submitted. Identifying lesions on the day of the procedure is an intermediate outcome. What really matters is does bowel prep (not dose of 1 type of bowel prep) impact cancer incidence and death. It is quite likely that the authors of the limited literature that will be reviewed did not measure this important outcome. That is fine but should be examined. The other outcomes should be specified using PICOTS. The authors plan to only include articles 2000 to present and say that this study is needed because the information requires synthesis independent of the more comprehensive Cochrane review. I am unclear how this article will add to previous reviews of PEG only prep such as a review of 12 trials of split dose in 2012 by Doug Rex. https://www.ncbi.nlm.nih.gov/pmc/articles/PMC3533212/ As this article says "However, any bowel preparation currently being used—including any type of 4-L or 2-L PEG preparation—can be split by administering half of it the evening before a colonoscopy and the other half the morning of a colonoscopy." Perhaps it isn't the dose but the timing of the dose that matters. Please comment. Perhaps if the authors read the old literature they might gain new insights on the mechanism that have been forgotten with time. I am curious why the Chinese literature was not included, only English language articles. Cochrane does not recommend restricting the review by any language. The authors stated that they updated their methods for consistency with the Cochrane guide. Please update this method as well or at least indicate how many articles of each language were excluded because a translator could not be found.
---

REVIEWER	Zhizheng Ge Renji Hospital, Shanghai Jiao Tong University School of Medicine, Shanghai Institute of Digestive Disease, Shanghai, China.
REVIEW RETURNED	11-Sep-2017

GENERAL COMMENTS	This revised manuscript refined the subject and method of this study, however narrowed the representative of this review. The manuscript has been greatly improved and is worthy of publication. Look forward to more well-designed articles.
---

VERSION 2 – AUTHOR RESPONSE

Comments from Reviewer #1 Professor Susan Hutfless (Johns Hopkins University):

Comment: The authors state that they revised the study to include cancer outcomes but no such changes exist in the track changed version of the draft submitted. Identifying lesions on the day of the procedure is an intermediate outcome. What really matters is does bowel prep (not dose of 1 type of bowel prep) impact cancer incidence and death. It is quite likely that the authors of the limited literature that will be reviewed did not measure this important outcome. That is fine but should be examined.

The other outcomes should be specified using PICOTS.

Response to comment: We'd like to extend our appreciation for your sparing time out of a busy schedule to review our manuscript and give this valuable evaluation. A point must be stated is that, in fact, we have included detection rate of polyp and adenoma according to the two peer reviewer's comments. However, colorectal cancer was not considered in previous revision version. Certainly, we are agreeing with peer reviewer's view which detection of lesions on the day of the procedure is an intermediate outcome. However, in clinical practice, good bowel preparation is mainly for better detection of lesions. And thus, bowel preparation efficacy is always evaluated according to the detection of gastrointestinal lesions. We consequently included the colorectal cancer to be as the outcome according to your comments (see line 132 and line 152 at page 7). Moreover, we also specified all outcomes of interest based on the PICOTS principle. But an explanation must be made is that all outcomes were evaluated based on all adult patients undergoing elective colonoscopy in endoscopy center, so we did not repeatedly introduce the patients and setting when defined each outcome. However, we have specified each outcome with the timing.

Comment: The authors plan to only include articles 2000 to present and say that this study is needed because the information requires synthesis independent of the more comprehensive Cochrane review. I am unclear how this article will add to previous reviews of PEG only prep such as a review of 12 trials of split dose in 2012 by Doug Rex. <https://www.ncbi.nlm.nih.gov/pmc/articles/PMC3533212/> As this article says "However, any bowel preparation currently being used—including any type of 4-L or 2-L PEG preparation—can be split by administering half of it the evening before a colonoscopy and the other half the morning of a colonoscopy." Perhaps it isn't the dose but the timing of the dose that matters. Please comment. Perhaps if the authors read the old literature they might gain new insights on the mechanism that have been forgotten with time.

Response to comment: Dear reviewer, it's our honor that you were invited to review our manuscript and warmly proposed these valuable comments. In the article entitled "Split Dosing for Bowel Preparation", Douglas K. Rex also stated that "there have not been many head-to-head comparisons between split-dose preparations". That is to say, the comparative efficacy between various split-dose preparations remains debate. More importantly, some meta-analyses have been designed to investigate the comparative efficacy between split- and same-day dose approach because of any bowel preparation currently being used can be split by administering half of it the evening before a colonoscopy and the other half the morning of a colonoscopy, but conclusive findings was also not generated. And thus, it is the urgent the time to adopt the network meta-analysis technique comprehensively investigating the comparative efficacy of all regimes and ranking all targeted bowel preparation regimes. However, the associated information cannot be obtained from the Cochrane review. We decided to only include articles published between 2000 and present according to Xie QS's article (retrieved from <http://pubmedcentralcanada.ca/pmcc/articles/PMC4047058/>).

Comment: I am curious why the Chinese literature was not included, only English language articles. Cochrane does not recommend restricting the review by any language. The authors stated that they updated their methods for consistency with the Cochrane guide. Please update this method as well or at least indicate how many articles of each language were excluded because a translator could not be found.

Response to comment: We would like to appreciate the reviewer for the constructive comments. As the peer reviewer stated, Cochrane recommend the reviewers to capture all potential articles under the permission of condition, for example. And thus, we have revised the Language Restriction according to this suggestion. (See line 134 to 136 at page 70)

Comments from Reviewer #2 Professor Zhizheng Ge (Renji Hospital):

Dear editor,

Comment: Thank you for inviting me to evaluate the article titled 'Effect of comparing 2 liters polyethylene glycol (PEG) alone or plus ascorbic acid and 4 liters PEG alone with each other for bowel preparation before colonoscopy: protocol for a systematic review and network meta-analysis' again. This revised manuscript refined the subject and method of this study, however narrowed the representative of this review. The manuscript has been greatly improved and is worthy of publication. Look forward to more well-designed articles.

Response to comment: Dear referee, we appreciate you for reviewing our manuscript and proposing pertinent comments.

We sincerely invite all previous reviewers to review our revised manuscript if it is convenient for them.

I'm looking forward to receiving reply on final decision of my manuscript from you and anonymous peer reviewers as soon as possible.

VERSION 3 – REVIEW

REVIEWER	Susan Hutfless Johns Hopkins University USA
REVIEW RETURNED	19-Sep-2017
GENERAL COMMENTS	The added text requires English language copy-editing. The editorial office may be able to provide this service.